# Tumor-Associated and Systemic Autoimmunity in Pre-Clinical Breast Cancer among Post-Menopausal Women

**DOI:** 10.3390/biom13111566

**Published:** 2023-10-24

**Authors:** Christine G. Parks, Lauren E. Wilson, Michela Capello, Kevin D. Deane, Samir M. Hanash

**Affiliations:** 1Epidemiology Branch, National Institute of Environmental Health Sciences, National Institutes of Health, Research Triangle Park, NC 27709, USA; 2Center for Population Health, Duke University School of Medicine, Durham, NC 27710, USA; 3Departments of Clinical Cancer Prevention, The University of Texas MD Anderson Cancer Center, Houston, TX 77030, USAshanash@mdanderson.org (S.M.H.); 4Division of Rheumatology, University of Colorado School of Medicine, Aurora, CO 80045, USA

**Keywords:** autoantibodies, breast cancer, epidemiology, immunology, biomarkers

## Abstract

Autoantibodies to tumor-associated antigens (anti-TAA) are potential biomarkers for breast cancer, but their relationship systemic autoimmunity as ascertained though antinuclear antibodies (ANA) is unknown and warrants consideration given the common occurrence of autoimmunity and autoimmune diseases among women. The relationship between anti-TAAs and ANA among women who were later diagnosed with breast cancer and others who remained cancer free in the Women’s Health Initiative cohort. The study sample included 145 post-menopausal women with baseline ANA data. A total of 37 ANA-positive women who developed breast cancer (i.e., cases; mean time to diagnosis 6.8 years [SE 3.9]) were matched to a random sample of 36 ANA-negative cases by age and time to diagnosis. An age-matched control sample was selected including 35 ANA-positive and 37 ANA-negative women who did not develop breast cancer (i.e., controls; follow-up time ~13 years [SE 3]). Baseline sera were assessed for Immunoglobulin G (IgG) antibodies, measured by custom microarray for 171 breast and other cancer-associated TAA. We used linear regression to estimate cross-sectional associations of ANA with log-transformed anti-TAA among cases and controls. Most anti-TAA did not vary by ANA status. Two anti-TAA were elevated in ANA-positive compared to ANA-negative cases: anti-PGM3 (*p* = 0.004) and anti-TTN (*p* = 0.005, especially in cases up to 7 years before diagnosis, *p* = 0.002). Anti-TAA antibodies were not generally related to ANA, a common marker of systemic autoimmunity. Associations of ANA with particular antigens inducing autoimmunity prior to breast cancer warrant further investigation.

## 1. Introduction

A growing body of research has sought to identify pre-clinical cancer-associated autoantibodies as potential biomarkers [1]. A diverse set of autoantibodies to tumor-associated antigens (anti-TAA) have been identified across a variety of cancers [2,3,4,5,6]. Tumor-specific autoantibodies could potentially be used to enhance the specificity of screening mammograms for breast cancer and reduce the burden of unnecessary biopsies [7,8]. A study of serum collected an average of 20 weeks prior to breast cancer diagnosis, among women who developed breast cancer in the Women’s Health Initiative (WHI), identified elevated serum IgG anti-TAA antibodies to more than 90 antigens [9]. Notably, some reactivity was observed for antigens typically associated with systemic autoimmune diseases. Thus, while anti-TAA antibodies are sensitive markers that may help identify the presence of early tumors [7,8,10,11], understanding their relationship with systemic autoimmunity is critical to determining their utility as screening biomarkers in the clinical setting.

Antinuclear antibodies (ANA) are a common marker of systemic autoimmunity found in 15–20% of women in the general population [12]. ANA are induced through an immune response to antigens released from apoptotic cells [13], with higher prevalence in parous women and older adults [12,14]. The gold standard assay for ANA (immunofluorescence on HEp-2 cells) detects IgG antibodies to over 150 known autoantigens, e.g., nucleic acids or other functional molecules, most of which are not associated with specific pathologies or clinical diseases. ANA and other more specific autoantibodies often accompany the development of autoimmune diseases [15], many of which are more common in women with autoimmune diseases, such as systemic lupus erythematosus and rheumatoid arthritis. Therefore, a better understanding of the relationship of autoimmunity with tumor-associated autoantibodies is warranted, given the common occurrence of ANA, autoimmune diseases, and breast cancer among women.

Using data and samples from the WHI, we explored the association between ANA and anti-TAA antibodies in baseline sera obtained from women who later developed breast cancer (i.e., “cases,” diagnosed an average of 7 years *after* serum collection), and in age and ANA-matched “controls” who did not develop breast cancer during follow-up. Serum specimens, obtained from a prior study of women with self-reported rheumatoid arthritis screened for ANA, were tested against a custom array of anti-TAA identified in prior studies [9,10,16,17,18,19,20,21].

## 2. Methods

### 2.1. Sample

The current study sample of 145 post-menopausal women was obtained from a subset of 9984 WHI participants with existing data on ANA, from a previous biomarker study of women who reported a diagnosis of rheumatoid arthritis (RA) at enrollment or follow-up (Ancillary Study BAA20, Appendix A) [22].

To evaluate the association of ANA with anti-TAA in women with a subsequent breast cancer diagnosis, we first identified a sample of 37 ANA-positive women who developed breast cancer for the first-time during follow-up, which was matched with a random sample of 36 ANA-negative women who developed breast cancer over the same follow-up time. Matching criteria included age (±2 years) and time to diagnosis (±1 year). Sample eligibility required complete data on hormone therapy, parity, estrogen-receptor tumor status and adequate serum volume (<200 μL). Incident breast cancer diagnoses were adjudicated on 96% of cases. Three cases with a positive ANA had a cancer history at baseline (1 lung, 1 endometrial/kidney, 1 lymphoma). Other cancers identified at follow-up (2 melanomas, 1 colon, 1 kidney) were equally distributed in ANA-positive and ANA-negative cases.

To evaluate the relationship of ANA with anti-TAA in women without known breast cancer, a random sample was selected of 35 ANA-positive (matched by level, as low, moderate or strong positives) and 37 ANA-negative women who did not develop breast or any other cancers during follow-up (i.e., controls), matched by age (±1 year) and in a time-forward manner, to ensure controls had at least as much follow-up time as cases.

The study was approved by IRB boards at the National Institute of Environmental Health Science and Fred Hutchinson Cancer Research Center. Existing samples and deidentified data were obtained from the WHI, which obtained informed consent from all participants.

### 2.2. Assays

Antinuclear antibodies (ANA) were measured on all specimens as previously described [22]. In brief, the samples were first screened to identify likely ANA-positives using the Bio-Rad Laboratories, Inc. Autoimmune EIA ANA Screening Test (Bio-Rad Laboratories, Inc., Hercules, CA, USA). The assay was designed to detect disease-associated antibodies [SSA (SSA 60 and SSA 52), SSB, anti-Sm, sm/RNP, RNP (RNP 68 and RNPA), anti-ribosomal protein, chromatin, anti-dsDNA, centromere, Scl-70, and Jo-1], with positivity determined based a level that was >95% sensitive for patients with systemic lupus erythematosus (SLE). These were subsequently confirmed by immunofluorescence (IFA) on Hep2 cells, with positives defined as having at least 2+ fluorescent intensity at a 1:320 serum dilution, rated as low moderate or strong. Specimens were also tested for rheumatoid factor (RF) by nephelometry (Dade Behring, Newark, DE, USA) and anti-CCP2 antibodies (Axis-Shield Diagnostics, Dundee, UK) [22].

Autoantibodies to tumor-associated antigens (anti-TAA) were assessed quantitatively using a custom array, including recombinant proteins that have previously elicited an autoantibody response to breast and other common cancers [9,10,16,17,18,19,20,21]. Specific TAA were chosen based on prior analyses of breast cancer and glycolysis pathways, while others were identified as non-specific tumor antigens or related to other solid tumors (colon, lung, prostate, and pancreas). Slides were spotted with 171 unique TAA antigens, including 2–3 replicates for 58 antigens (34%). Recombinant protein antigens (Life Technologies, Carlsbad, CA, USA) were suspended in printing buffer (250 mmol/L of Tris-HCl, pH 6.8, 0.5% sodium dodecyl sulfate, 25% glycerol, 0.05% TritonX-100, 75 mmol/L of dithiothreitol) and spotted at 1 mg/mL onto 16-pad nitrocellulose-coated slides (Maine Manufacturing, Kennebunk, ME, USA) using a contact printer (Genetix QArray2). The printing buffer was included on the slides as negative control. Purified human immunoglobulin G (IgG) and EBNA (Epstein–Barr Virus Nuclear Antigen 1) were used as positive controls.

Sixteen samples were hybridized on each slide at a dilution of 1:150, and a secondary antibody reaction with a fluorescent anti-human IgG (1 μg/mL, DyLight 649-conjugated rabbit anti-human IgG, Jackson ImmunoResearch) was performed. Then, the protein arrays were scanned with a GenePix scanner (Molecular Devices GenePix Pro 6.1 4000B, San Jose, CA, USA) using a red laser (635 nm), and local background-subtracted median spot intensities were generated using GenePix Pro 7 Software. Normalized values subtracted the log transformed anti-TAA (Log2TAA) median array intensities from individual intensities (median centered normalization) to account for variability between arrays. Valid levels were detected for most antigens, with low detects reflected by a Log2TAA value of zero. There was no significant difference in zero values by case and ANA status (Appendix A). Correlations of positive controls (anti-IgG and EBNA) were strong across two plates (rho = 0.82). On a third of the antigens with replicates, cross-array variability ranged from rho = 0.77 to 0.86.

### 2.3. Analysis

Differences in anti-TAA by ANA status were examined separately in women diagnosed with breast cancer during follow-up (cases) and in women who were not diagnosed with breast or other cancers during the same time (controls). Initial descriptive analyses visually explored the average of median Log2TAA values for each woman, stratified by ANA and case status. Ratio values comparing median anti-TAA values in ANA-positive versus ANA-negative women, stratified by case status, were calculated, and screened against the null (i.e., 1.0, indicating equal intensities) using a *t*-test *p*-value of 0.05 for the mean value and a median ratio value of <0.90 or >1.10. A total list of all anti-TAA means ratios, *p*-values, and median ratios, are shown in Appendix A, and the summary distribution was plotted for visual inspection. For Log2TAA ratios exceeding this threshold in one or more groups, mean Log2TAA values and standard deviations are shown in Appendix A, along with Persons’ correlation statistics with age, time to diagnosis (in cases), EBNA and the other anti-TAAs meeting screening criteria. For interpretation of results from screening analysis, a Bonferroni threshold of *p* = 0.0058 for statistical significance was determined by the number of total comparisons (0.05 divided by 171).

Using linear regression, we examined ANA as a predictor of anti-TAA intensities for those antigens that met the screening threshold (SAS Version 9.1, Cary, NC, USA). Models were run separately in controls and cases, and in cases, models were also stratified by time from serum collection to diagnosis (<7 vs. 7+ years) in cases. We first adjusted for age (Model 1) and then added EBNA, RF, and current hormone replacement therapy (Model 2; presented in results). Sensitivity analyses excluded women who were currently using DMARDS, cases with a history of other cancer at enrollment or during follow-up, or reported a diagnosis of systemic lupus erythematosus, and limited to women ER-positive cancers. Because age is an important determinant of both ANA and cancer risk, we explored the consistency of our main findings in models stratified by age (younger than 65 years and 65 and older). Zero values were included in regression models of continuously distributed Log2TAA. In sensitivity analyses, we imputed values as half the lowest detected value. We did not test for statistical interactions in these exploratory models.

## 3. Results

Table 1 shows sample characteristics, including balanced follow-up time by ANA status in cases (median 7 years from serum collection to breast cancer diagnosis) and controls (median 13 years follow-up). Compared with ANA-negative cases, ANA-positive cases were more likely to be RF-positive and take disease modifying anti-rheumatic drugs (DMARDs) at baseline. No differences by ANA status were seen among controls.

Visual inspection of overall anti-TAA antibody reactivity (depicted as the median Log2TAA across all 171 antigens per woman) showed a broad distributed that did not appear to vary by ANA status in either cases or controls (Figure 1). Comparing the median intensity of the 171 individual antigens by ANA status showed no difference in the ratio of median values, though we noted a broader distribution of ratio values and more high outliers among cases, indicating more instances of elevated anti-TAA among ANA+ versus ANA-women (Figure 2).

Of all anti-TAA screened (listed in Appendix A), only eight Log2TAA ratios were found to deviate from the null (*p* < 0.05; Table 2). Five specific anti-TAA antibodies were elevated comparing the ratio of Log2TAA intensities (i.e., anti-TAA levels) in ANA-positive to ANA-negative women. In cases, anti-TTA levels were significantly higher based on the threshold adjusting for multiple comparisons for two antigens: PGM3, *p* = 0.004; TTN, *p* = 0.005. Similar, non-significant elevations of these anti-TAA with ANA were seen in controls. Two other anti-TAA were higher in ANA-positive women (DUSP26 in cases and PRKD1 and UCLH1 in controls) and three were lower (ACTA2, C5orf45 and C13orf24), only in cases.

Across these eight anti-TAA, we examined correlations with age and time to diagnosis (Table 3). We saw no correlation of age with anti-TAA in cases, regardless of ANA status. In controls, however, levels of two anti-TAA were significantly correlated with older age (PGM3 and UCHL1). In ANA-positive cases, time to breast cancer diagnosis decreased in association with PGM3 intensity and were increased for DUSP26. Conversely, levels of PRKD-1 were associated with longer time to diagnosis in ANA-negative women. Correlations among anti-TAA and with EBV reactivity are shown in Appendix A. Given suggestive correlations of some anti-TAA with days to diagnosis, we also describe the distribution of values stratified by median time to diagnosis (7 years) in Appendix A.

Given these observed differences in correlations (in Table 3), we modeled ANA as a predictor of Log2TAA levels in linear regression models stratified by case status, and by time to diagnosis in cases, adjusting for age. We also ran a multivariable model additionally adjusting for anti-EBNA levels, RF and hormone replacement therapy (Table 4). Only one association met the Bonferroni-threshold for statistical significances: i.e., in cases with a shorter time to diagnosis (<7 years), having ANA antibodies was positively and significantly associated with TTN anti-TAA in both models (age-adjusted beta = 1.54, *p* = 0.004, multivariable beta = 1.63, *p* = 0.002). In age-stratified models, the association of TTN levels with ANA was most apparent in younger (<age 65 years; age-adjusted beta = 2.6, *p* = 0.004) but not older cases (beta = 1.08; *p* = 0.15). A sensitivity analysis with imputed values for two Log2TAA zero values (PRKD1 and UCHL1) showed no substantial differences (Appendix A). Results were not substantively changed after excluding women using DMARDs, reporting systemic lupus erythematosus, with prior cancer history, or when limited to ER-positive cases (Appendix A).

## 4. Discussion

Given the high prevalence of systemic autoimmunity in the general population, especially in women and older adults, the relationship of systemic and tumor-associated autoimmunity is an important consideration in determining the utility of tumor-specific antibodies as biomarkers in clinical screening and cancer research. We found that having a positive ANA was not broadly associated with TAA across an array of 171 antigens selected, based on prior evidence of associations with breast and other cancers. While largely null, our findings indicate a need for further investigation of TAA and ANA prior to breast cancer diagnosis. The concept of ANA as a potential confounder assumes a non-causal relationship of ANA with both anti-TAA (e.g., in women without cancer) and with breast cancer. Although a few differences in specific anti-TAA levels were seen, depending on ANA status among controls (PRKD1 and UHL1), no significant differences remained after adjusting for multiple comparisons. Among cases, however, significantly higher levels of two specific anti-TAA (PGM3 and TTN) were observed in ANA-positive compared to ANA-negative women (Table 2), and in covariate-adjusted models, significantly higher anti-TTN levels were seen in women diagnosed with breast cancer within seven years (Table 4).

The association of ANA with anti-TTN TAA levels was stronger in women whose cancers were diagnosed within seven years, which suggests a temporal relationship of these autoantibodies in the development of breast cancer. Titin (encoded by the TTN gene) is a very large protein, which plays an important structural, developmental and regulatory role in skeletal and cardiac muscle [23,24]. Notably, the gene has also been identified as a key tumor driver, with functional germline mutations associated with breast cancer [25,26]. Literature on autoimmunity to titin is limited. A growing body of research shows antibodies to titin can be associated with the autoimmune muscular disease, myasthenia gravis (closely associated with thyoma) [27], while a study of 44 cancer patients with peripheral nervous system paraneoplastic syndromes reported elevated levels of anti-TTN in two patients (one of whom had breast cancer) [28,29]. Against this background, our findings suggest a need for more research on the relationship of TTN with both cancer and autoimmunity.

Our initial screening comparisons also showed that ANA was positively associated with anti-PGM3 TAA levels in cases, and an elevated but non-statistically significant association was seen for anti-PGM3 and ANA in controls. Subsequent adjusted models suggested modest associations of similar strength in both cases and controls. Phosphoacetylglucosamine mutase (PGM3) is one of several paralogues (PGM1-3,5) that control glucose metabolism in most cells [30], and defects in PGM3 have been linked to autoimmunity [31]. Our findings may suggest a more general relationship of ANA with anti-PGM3 TAA regardless of breast cancer, which could have implications for use of this antigen as a potential cancer biomarker.

In our overall analyses, no other associations reached statistical significance at the Bonferroni threshold of *p* = 0.0058; however, some findings were suggestive. For example, ANA was associated with elevated levels of anti-PRKD1 TAA in controls, but negatively associated in cases at least 7 years prior to diagnosis. Protein Kinase D1 (PRKD1) is a serine/threonine kinase expressed in normal breast tissue, where it is responsible for maintaining the epithelial phenotype [32]. Methylation of the PRKD1 promoter has been linked to silencing seen in breast cancer tumor progression, and targeted upregulation of PRKD1 has been suggested as an approach to reduce breast cancer invasiveness [32]. It is plausible that an autoimmune predisposition could be related to developing both ANA and antibodies to PRKD1 in women with a pattern of normal expression, but reasons for a negative association in women who developed cancer are less apparent, especially for such distal diagnoses.

Prior studies suggested levels of breast cancer-specific anti-TAA may be lower in current users of hormone therapy, especially estrogen [16], but we observed no confounding with current hormone use and our findings were similar when limited to ER-positive breast cancer cases. Rheumatoid factor (RF) has been commonly associated with RA, but these antibodies to the Fc region of immunoglobulins occur across a spectrum of other autoimmune conditions, infections, and sometimes in healthy individuals [33]. In the source population from which our sample was derived, RF was significantly higher in women with ANA antibodies (Appendix A), but in our sample, this difference was only significant in women who became cases and not in controls (Table 1). We are aware of only one previous study showing elevated RF in breast cancer cases [34]. Further studies are warranted.

In addition to the example described for TTN, myasthenia gravis, and thyoma described above, cancer-associated autoimmunity has been found to occur across a range of diseases, such as myositis and scleroderma, considered to be a paraneoplastic phenomenon [35]. Recent studies have also shown an association between mutations in the RNA polymerase, POLR3A, with development of a specific autoantibody in scleroderma patients with cancer (anti-RNA polymerase I/III antibodies) [36], suggesting that pathogenic autoimmunity among some scleroderma cases may arise from an immune response to mutated tumor antigens.

Cancer immunoediting has been proposed to play a role in immune surveillance and control of growing tumors [37]. In the present study, the association of ANA with anti-TTN TAA antibodies was only seen in cases within 7 years of diagnosis. Antibodies measured closer to diagnosis may reflect an early tumor-specific immune response that is more robust in women with a positive ANA, which may therefore be a marker of their potential to mount an autoimmune response. Alternatively, ANA could be a byproduct of tumor-specific autoimmunity. The idea that preclinical autoimmunity could play a functional role against cancer seems somewhat provocative, but the growing use of immunotherapies proves the efficacy of harnessing the immune system to control established tumors. A growing body of research also suggests potential protective associations of RA and SLE with risk of breast cancer, though the mechanisms are unknown, and the evidence is inconsistent [38,39,40]. The current study design and relatively small sample size precludes a definitive evaluation of ANA and autoimmune diseases in relation to breast cancer, though we note a lower frequency of self-reported lupus in cases compared to controls, regardless of ANA status (Table 1).

Antinuclear autoantibodies associated with clinical autoimmune diseases may have different characteristics, causes and consequences, compared to those seen in the general population. Autoimmune patients tend to have higher titers or disease-related autoantibodies to complexes of proteins with DNA or RNA [41], and are less likely to have antibodies to the DFS70/LEDGFp75 antigen (common in non-autoimmune patients) [42]. The causes and implications of ANA in the general population are not well understood; ANA prevalence is higher in women and in aging adults [12,14]. Although Immunoglobulin (Ig)G antibodies are often thought to reflect potential for pathologic autoimmune responses, ANA-IgG in healthy individuals may reflect other processes, and ANA may even play a physiologic role in clearing cellular debris [43]. Age is an important determinant of both cancer risk and ANA, and future studies should consider the influence of immune aging, which has implications for interpreting the association between systemic and cancer-specific autoimmunity. Interestingly, the association of ANA with anti-TTN in cases (within 7 years of diagnosis) was most apparent in younger women (<67 years). This could be due to many factors, including a less robust immune response at older ages. The natural history of ANA outside of clinical settings is not well understood; thus, these cross-sectional associations of ANA with TAA warrant cautious interpretation and replication in a larger sample and including repeated specimens (if possible) and more proximal sampling (e.g., <2 years) prior to breast cancer diagnosis.

This study has limitations, especially given the relatively small sample, which limited sub-analyses. However, the focused design allowed us to address our research question overall and in the context of future breast cancer. Our TAA assay included many specific antigens, increasing the number of comparisons and the possibility that associations could arise due to chance. Therefore, we highlighted only those results that would have been statistically significant at a *p*-value adjusted for multiple comparisons (0.0058). Because analyses of smaller sub-samples can introduce bias through effect size magnification, we did not consider all 171 anti-TAA with respect to age or other potential modifiers. Our results support the development of larger studies to provide adequate power for stratified analyses.

Our study sample was drawn from a larger sample of women who reported rheumatoid arthritis (RA) who had existing data on ANA. The proportion of clinical RA and lupus in our study sample is likely to be low, given the poor specificity of self-report [44,45] and low prevalence of DMARD use and anti-CCP antibodies. To maximize the number of ANA-positive breast cancer cases, our sample included a small number of women with a history of another cancer prior to specimen collection or during follow-up. Sensitivity analyses yielded no substantial differences in TAA/ANA associations after excluding women with DMARDs, CCP antibodies, other cancer, or reported cases of systemic lupus erythematosus. The initial screening ELISA focused on a limited panel of disease-specific autoantigens, with subsequent results confirmed by HEP-2 assays at a level consistent with uses in clinical settings. While 15% of the screened sample was identified as ANA-positive, the assay has low sensitivity for the full range of antinuclear antigens, most of which are not associated with specific pathologies or clinical diseases. Thus, the true prevalence of ANA may have been higher and represented different specificities, resulting in misclassified ANA-negative women in our analyses. Future research on ANA and anti-TAA would benefit from an initial screen by the HEP-2 assay, while also including data on ANA specificities, titers, and staining patterns to contextualize the results and better understand underlying relationships driving any observed associations.

The current study was not intended to investigate the utility of anti-TAA antibodies in breast cancer screening, given the extended time between serum collection and diagnosis (average of 7 years). Results may have been different for samples taken closer to the time of diagnosis. While there is some evidence of an association of ANA with breast cancer in clinical studies (e.g., as reviewed by Nisihara et al. [46]), this question has been infrequently studied, and we are unaware of prospective research on this question in the general population. Anti-nuclear antibodies, with the dense fine speckled pattern targeting the LEDGFp75 (lense epitheluium growth factor p75) antigen, are one of the most common specific ANA in non-autoimmune patients [47], and have been characterized as an “epigenetic reader” that may be upregulated as a stress protein in cancer and inflammatory conditions [42,48,49]. The prevalence of this antibody and other non-disease associated ANA may be present among those failed the initial screening by ELISA. Controls were selected excluding all cancer types, whereas three ANA-positive cases had a history of cancer at enrollment. Results were unchanged with these that were excluded from the sample.

In conclusion, our findings that ANA positivity was not broadly associated with higher TAA reactivity suggest that ANA are unlikely to confound studies of tumor autoantibodies and cancer. However, observed associations of ANA with anti-TAA in women who later developed cancer suggest that cancer-associated adaptive immune responses may exist to years prior to the development of clinically apparent tumors, and support the development of larger prospective studies with repeated measures to investigate autoimmunity and pre-clinical cancer.

## Figures and Tables

**Figure 1 biomolecules-13-01566-f001:**
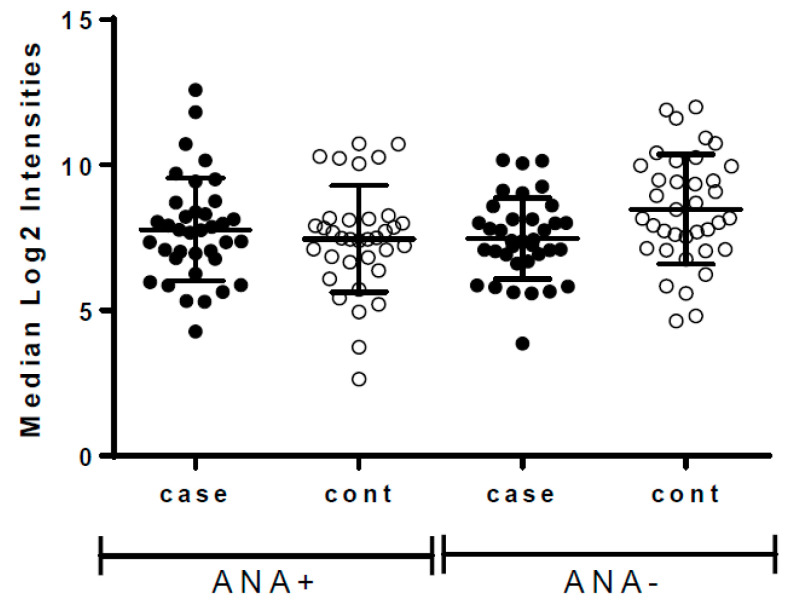
Median Log2TAA intensity (i.e., anti-TAA levels) among women who were positive for anti-nuclear antibodies (ANA+) and those who were negative (ANA−), stratified by whether they later developed breast cancer (cases, N = 37 ANA+ and 36 ANA−) or did not (controls, N = 35 ANA+ and 37 ANA−). Bars indicate the median and interquartile range across women in each group.

**Figure 2 biomolecules-13-01566-f002:**
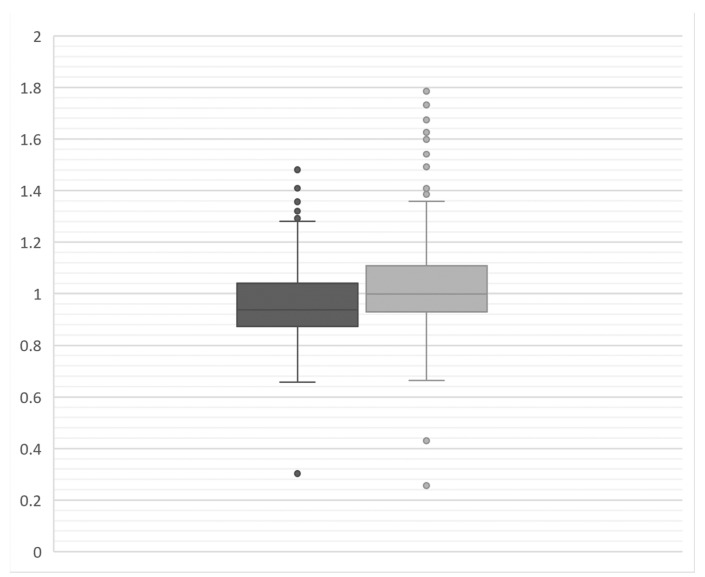
Ratio of 171 total Log2TAA values comparing ANA-positive (ANA+) to ANA-negative (ANA−) women, among cases (N = 37 ANA+ and 36 ANA−, shown in light grey) and controls (N = 35 ANA+ and 37 ANA−, shown in darker grey). A ratio value of 1.0 shows no difference in intensity by ANA status, while ratios >1.0 suggest greater reactivity among ANA+ versus ANA− women. A box and whisker plot shows the median values, interquartile range, and 95% confidence intervals.

**Table 1 biomolecules-13-01566-t001:** Characteristics by baseline ANA status in incident breast cancer cases and controls.

	Incident Breast Cancer			Controls (No Breast Cancer during Follow-Up)		
	ANA-Pos N = 37	ANA-Neg N = 36	*p*-Value ^a^	ANA-Pos N = 35	ANA-Neg N = 37	*p*-Value ^a^
Matching factors						
Age—years (SE)	64.9 (6.0)	64.9 (5.7)	*---*	64.7 (6.1)	64.9 (6.0)	*---*
Years to breast cancer or follow-up time (SE)	6.8 (3.9)	6.9 (3.8)	*---*	13.1 (3.6)	13.0 (3.0)	*---*
Other autoantibodies, medications, and hormone use	N (%)	N (%)		N (%)	N (%)	
Anti-CCP positive	3 (8)	1 (3)	0.32	4 (11)	4 (11)	0.97
Anti-RF positive	10 (27)	3 (8)	0.038	7 (19)	4 (11)	0.31
Current DMARD use	4 (11)	0 (--)	**0.044**	1 (3)	3 (8)	0.32
Current Prednisone	1 (3)	2 (6)	0.54	0 (--)	2 (5)	0.16
Current estrogen use E only E + P	11 (30) 4 (12)	8 (22) 5 (14)	0.56	7 (19) 5 (13)	10 (27) 3 (8)	0.64
Other Characteristics						
Lupus (self-reported at baseline or follow-up)	1 (3)	2 (5)	0.57	6 (17)	6 (17)	0.96
Breast cancer tumor ER positive Invasive	27 (87) ^b^ 28 (76)	27 (75) 31 (86)	0.22 0.26	NA NA	NA NA	---

ANA, Antinuclear Antibodies. Bold = *p* < 0.05. ^a^ Comparing ANA-positive with ANA-negative women in cases and controls separately. ^b^ Excluding 6 missing or do not know ER status (all ANA-positive).

**Table 2 biomolecules-13-01566-t002:** Higher or lower anti-TAA levels shown as the ratio of Log2TAA intensities in ANA+ compared to ANA- women in cases and controls.

	Cases (N = 37 ANA+, 36 ANA−)		Controls (N = 35 ANA+, 37 ANA−)		
Antigen	Median Ratio ^a^	*p*-Value ^b^	Median Ratio ^a^	*p*-Value ^b^	Function/Pathways
**Higher anti-TAA in ANA-positive versus ANA-negative women**					
PGM3	**1.48**	**0.004**	1.67	0.128	Phosphoacetylglucosamine mutase: mediates glycogen formation/utilization, defects linked to autoimmunity
TTN	**1.41**	**0.005**	1.79	0.129	Tintin: protein kinase, widespread in muscle; antibodies in scleroderma; somatic mutation in cancer
DUSP26	**1.17**	**0.018**	1.15	0.492	Dual specificity phosphatase 26: protein-tyrosine-phosphatase
PRKD1	1.05	0.801	**1.73**	**0.024**	Protein Kinase D1: rap1 signaling pathway.
UCHL1	0.89	0.970	**1.33**	**0.046**	Truncated calcium binding protein
**Lower TAA in ANA-positive versus ANA-negative women**					
ACTA2	**0.86**	**0.021**	0.88	0.629	Actin Alpha 2, smooth muscle, aorta; interacting with TTN/TNF; exosomal protein of colorectal cancer
C5orf45	**0.79**	**0.032**	0.81	0.397	Chromosome 5, open reading frame 45; expressed in breast cancer MCF7, chronic B-lymphocytic leukemia
C13orf24	**0.84**	**0.008**	0.92	0.994	Progesterone-induced blocking factor 1: immunomodulatory, increases TH2 response, NK cells

ANA, Antinuclear Antibodies; ANA+, ANA positivity. Case = women who later developed breast cancer, and Control = women who did not. ^a^ Median ratio of Log2TAA antibody levels: ANA-positive versus ANA-negative in cases and controls, shown in bold if *p* < 0.05, comparing to the null value of 1.0 (i.e., no difference). A value above 1 indicates higher Log2TAA intensity in ANA+ versus ANA− women (i.e., >1.0), whereas a value less than 1 indicates lower levels among ANA+ women. Those with Log2TAA levels = 0 were excluded from ratio calculations; this included 13 cases (6 ANA+ and 7 ANA−) and 14 controls (4 ANA+ and 10 ANA-) for PRKD1, and 7 ANA− cases and 2 ANA− controls for UCHL1. ^b^ Unadjusted *p*-value for *t*-test comparing mean ratio Log2TAA to the null (1.0).

**Table 3 biomolecules-13-01566-t003:** Levels and correlations of Log2TAA, stratified by ANA status, with age among cases and non-cases, and with days to breast cancer diagnosis in cases ^a^.

TAA	Log2TAA Mean (SE)	Range	Age in Years (SE)	Days to dx (SE)
**ANA-positive breast cancer cases (N = 37)**				
			64.9 (6.0)	2484 (1396)
EBNA	13.9 (1.7)	10–17	−0.34	−0.27
PGM3	8.5 (0.86)	6.7–10	−0.02	**−0.38**
TTN	8.5 (1.4)	6.0–13	0.09	−0.15
DUSP26	9.9 (0.88)	7.3–12	0.13	−0.30
PRKD1	4.5 (2.3)	0–7.7	0.21	0.19
UCHL1	9.4 (1.2)	6.2–13	0.17	−0.01
ACTA2	12 (0.5)	11–13	−0.09	0.04
C5orf45	11 (0.7)	9.9–15	0.30	0.14
C13orf24	11 (0.41)	10–12	0.24	0.03
**ANA-negative breast cancer cases (N = 36)**				
			64.9 (5.7)	2504 (1404)
EBNA	14.1 (1.7)	11–18	−0.13	−0.22
PGM3	8.2 (0.7)	6.8–10	−0.11	−0.09
TTN	8.0 (1.1)	6.0–11	−0.14	0.30
DUSP26	9.6 (1.0)	7.3–11	0.10	**−0.35**
PRKD1	4.4 (2.4)	0–7.7	−0.06	**0.50**
UCHL1	9.4 (1.3)	6.2–13	−0.20	0.16
ACTA2	12 (0.4)	11–13	−0.21	−0.04
C5orf45	11 (0.8)	10–15	−0.01	0.22
C13orf24	11 (0.7)	10–13	0.14	−0.01
**ANA-positive controls (N = 35)**				
			64.8 (6.0)	NA
EBNA	13.9 (1.7)	11–17	−0.28	---
PGM3	8.6 (1.5)	5.0–12	**0.41**	---
TTN	8.1 (1.2)	5.3–12	−0.00	---
DUSP26	9.7 (1.1)	7.4–11	0.01	---
PRKD1	4.5 (2.5)	0–8.1	0.24	---
UCHL1	9.5 (2.4)	0–15	**0.38**	---
ACTA2	12 (0.5)	10–13	0.24	---
C5orf45	11 (0.6)	9.7–12	0.24	---
C13orf24	11 (0.6)	10–14	0.14	
**ANA-negative controls (N = 37)**				
			64.9 (6.0)	NA
EBNA	13.7 (1.8)	11–16	−0.04	---
PGM3	8.3 (1.7)	5.0–12	0.00	---
TTN	7.9 (1.3)	5.3–12	0.25	---
DUSP26	9.7 (1.2)	7.4–11	−0.11	---
PRKD1	3.9 (2.6)	0–7.2	−0.02	---
UCHL1	8.9 (2.8)	0–13	0.22	---
ACTA2	12 (0.6)	10–13	0.05	---
C5orf45	11 (0.6)	9.7–12	0.14	---
C13orf24	11 (0.5)	9.7–12	−0.09	

ANA, Antinuclear Antibodies; EBNA (Epstein–Barr Virus. Case = women who later developed breast cancer, and Control = women who did not. Bold = *p* < 0.05. ^a^ Those with Log2TAA levels = 0 were excluded from ratio calculations: 13 cases (6 ANA+ and 7 ANA−) and 14 controls (4 ANA+ and 10 ANA−) for PRKD1, and 2 ANA− controls for UCHL1.

**Table 4 biomolecules-13-01566-t004:** Linear regression models showing the covariate-adjusted difference in Log2TAA levels associated with ANA-positivity among controls and among cases, stratified by time to diagnosis ^a^.

	Controls	Pre-Clinical Breast Cancer Cases	
Antigens	No Breast Cancer Diagnosis N = 64 Beta (*p*-Value)	7+ Years to Diagnosis N = 30 Beta (*p*-Value)	<7 Years to Diagnosis N = 35 Beta (*p*-Value)
PGM3	0.64 (0.09)	0.42 (0.12)	0.56 (0.09)
TTN	0.44 (0.16)	0.11 (0.84)	**1.63 (0.002)**
DUSP26	0.20 (0.49)	0.80 (0.07)	0.37 (0.12)
PRKD1	**1.36 (0.025)**	**−1.53 (0.044)**	0.85 (0.38)
UCHL1	**1.25 (0.032)**	−0.25 (0.58)	−0.02 (0.97)
ACTA2	−0.13 (0.28)	−0.13 (0.48)	**−0.26 (0.041)**
C5orf45	−0.18 (0.15)	−0.52 (0.14)	−0.30 (0.031)
C13orf24	−0.01 (0.97)	−0.30 (0.14)	−0.32 (0.09)

ANA, Antinuclear Antibodies. Case = women who developed breast cancer during follow-up, and Control = women who did not. Bold = *p* < 0.05. ^a^ Betas, calculated through linear regression, show the difference per unit change in Log2TAA values for ANA+ versus ANA− women, adjusting for age, EBNA antibody titers, RF positivity, and current use of hormone therapy. A positive Beta value indicates higher Log2TAA intensity among ANA+ versus women (i.e., greater than zero), whereas a negative Beta value indicates lower Log2TAA intensity among ANA+ women. Sample size reduced due to missing covariate data in 7 cases and 8 controls.

## Data Availability

Data and analytic code for replication of the paper may be requested from the authors, pending approval of the WHI parent study.

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
