# Peer review of "Tumor-Associated and Systemic Autoimmunity in Pre-Clinical Breast Cancer among Post-Menopausal Women"

_biomolecules, 2023, doi:10.3390/biom13111566_

Round 1

Reviewer 1 Report

Overall, this is an interesting topic and the experiments are well designed.  However, there are several issues for the authors to address prior to acceptance. 

1. The authors used a customized 171-antigen array, please explain how the 171 antigens were selected and why.

2. Please add a Table to include the autoantibody results from the 171-antigen array, either in the main document or supplementary data, so that the readers can take an overall look at what are the levels of the other autoantibodies aside from the ones presented here. 

3.  In Figure 1, regarding the ratio of anti-TAA levels, it is unclear which anti-TAA was measured and calculated, the total anti-TAA? if so, how? please also include the sample size and statistics used for Figure 1. Also, please describe how the ratio of anti-TAA levels (positive vs negative) was calculated.

4.  Please indicate the sample size and statistics/methods used for Table 2.

5. Please indicate the sample size for Table 3.

6. Please indicate the values of anti-TAA in Table 3 and Table 4.

7. In your anti-TAA antigen array results, please indicate the ranges for anti-TAA "positivity". If you use fluorescence intensity or log-transformed FI, what are the ranges for a valid value of anti-TAA?  

8. If possible, please use ELISA to valid at least the anti-TTN and anti-PGM3 results.

9. Please clarify what are the association statistic values in Table 4, why, and how some of them are greater than 1?

Author Response

Reviewer 1

Overall, this is an interesting topic and the experiments are well designed.  However, there are several issues for the authors to address prior to acceptance. 

Thank you for your careful review, thoughtful questions, and suggestions, which have helped to improve the clarity and quality of the paper. In our responses, we have added more description of the anti-TAA included in the study and methods used to evaluate possible differences by ANA status. Changes also include a new figure (replacing Figure 1), greater representation of the underlying data on Log2TAA used in ratio calculations and regression modeling, and new supplemental materials detailed below.

  1. The authors used a customized 171-antigen array, please explain how the 171 antigens were selected and why.

The choice of antigens for the array was informed by prior work by the senior author (Dr. Hanash). This was previously included in the Methods section (Page 5), but we have added some of this information and citations to the introduction. The associated genes are listed in Supplemental Table 2 (see response to second comment).

P4 (lines 68-70): “Serum specimens, obtained from a prior study of women with self-reported rheumatoid arthritis screened for ANA, were tested against a custom array of anti-TAA identified in prior studies [9, 10, 16-21].”   

  1. Please add a Table to include the autoantibody results from the 171-antigen array, either in the main document or supplementary data, so that the readers can take an overall look at what are the levels of the other autoantibodies aside from the ones presented here. 

We have added a table to the supplemental materials (Supplemental Table 2) including the complete list of Log2TAA tested and the comparative ratios for levels detected in women with ANA versus those without, stratified by case status. The tables also includes ratios for controls (EBNA and IgG1), as well as the screening p-values from which Table 2 values were derived.

P6-7 (lines 135-140): “Initial descriptive analyses visually explored the average of median Log2TAA values for each woman, stratified by ANA and case status. Ratio values comparing anti-TAA values in ANA-positive versus ANA-negative women, stratified by case status, were calculated, and screened against the null (i.e., 1.0, indicating equal intensities) using a t-test p-value of 0.05 for the mean value and a median ratio value of <0.90 or >1.10. A total list of all anti-TAA means ratios and p-values, along with median ratios, are shown in Supplemental Table 2, and the summary distribution was plotted for visual inspection.”

Example of Supplemental Table 2: “Supplemental Table 2. Screening Log2TAA ratios comparing values in ANA-positive with ANA-negative women, by case status”

Cases

Controls

Gene Symbol

p-valuea

Meanb

Median

p-valuea

Meanb

Median

AADAT

0.141

0.88

0.87

0.961

1.00

1.04

ABCA8

0.057

1.63

1.29

0.243

1.18

1.22

  1. In Figure 1, regarding the ratio of anti-TAA levels, it is unclear which anti-TAA was measured and calculated, the total anti-TAA? if so, how? please also include the sample size and statistics used for Figure 1. Also, please describe how the ratio of anti-TAA levels (positive vs negative) was calculated.

The original Figure 1 was intended to graphically address whether distribution of the total number of the total number of anti-TAA levels, i.e., Log2TAA intensities, were greater among ANA+ versus ANA- women, and to show how these performed relative to case status. We realize that this leaves out an important conceptual step showing the absolute values, so we added a new Figure 1 and renamed this as Figure 2 with a more extensive heading addressing the questions above. The Statistical testing was not performed testing for differences in these distributions by case status. We updated the methods to explain the derivation and interpretation of the Log2TAA ratio values used for comparisons. See response to question 2. No statistical testing was performed on these summary distributions.

 P9 (line 176-180): “Figure 1. Median Log2TAA intensity (i.e., anti-TAA levels) among women who were positive for anti-nuclear antibodies (ANA+) and those who were negative (ANA-), stratified by whether they later developed breast cancer (cases, N=37 ANA+ and 36 ANA-) or did not (controls, N=35 ANA+ and 37 ANA-). Bars indicate the median and interquartile range across women in each group.”

Page 10 (line 187): “Figure 2. Ratio of 171 total Log2TAA values comparing ANA-positive (ANA+) to ANA-negative (ANA-) women, among cases (N=37 ANA+ and 36 ANA-, shown in light grey) and controls (N=35 ANA+ and 37 ANA-, shown in darker grey). A ratio value of 1.0 shows no difference in intensity by ANA status, while ratios >1.0 suggest greater reactivity among ANA+ versus ANA- women. A box and whisker plot shows the median values, interquartile range, and 95% confidence intervals.”

P9 (lines 169-174):” Visual inspection of overall anti-TAA antibody reactivity (depicted as the median Log2TAA across all 171 antigens per woman) showed a broad distributed that did not appear to vary by ANA status in either cases or controls (Figure 1). Comparing the median intensity of the 171 individual antigens by ANA status showed no difference in the ratio of median values, thought we noted a broader distribution of ratio values and more high outliers among cases, indicating more instances of elevated anti-TAA among ANA+ versus ANA- women (Figure 2).”

  1. Please indicate the sample size and statistics/methods used for Table 2.

The table title has been edited to better describe the methods used to derive values in the table. Column headings now include the sample sizes, and statistical testing is described in a new footnote (b) as indicated below.

P10 (line 189): “Table 2. Higher or lower anti-TAA levels shown as the ratio of Log2TAA intensities in ANA+ compared to ANA- women in cases and controls.”

Cases

(N=37 ANA+, 36 ANA-)

Controls

(N=35 ANA+, 37 ANA-)

Antigen

Median ratioa

P-valueb

Median ratioa

P-valueb

Function / pathways

P11 (line 219): “bUnadjusted p-value for t-test comparing mean ratio Log2TAA to the null (1.0). 

P10 (line 191): “Of all anti-TAA screened (listed in Supplemental Table 2), only 8 Log2TAA ratios were found to deviate from the null (p<0.05, Table 2).”

  1. Please indicate the sample size for Table 3.

Sample size has been added to the column headers of Table 3 (see example below). Note – this table was reorganized to integrate requested values (comment 6).

P13 (line 223): “Table 3. Levels and correlations of Log2TAA, stratified by ANA status, with age among cases and non-cases, and with days to breast cancer diagnosis in casesa

TAA

Log2TAA Mean (SE)

Range

Age in years (SE)

Days to dx (SE)

ANA-positive breast cancer cases (N=37)

64.9 (6.0)

2484 (1396)

EBNA

13.9 (1.7)

10-17

-0.34

-0.27

PGM3

8.5 (0.86)

6.7-10

-0.02

-0.38

  1. Please indicate the values of anti-TAA in Table 3 and Table 4.

Please see changes to Table 3 illustrated above, including include Log2TAA values previously only shown in the supplemental materials (Supplemental Table 2.) These summary values (mean, SD) represent the values of Log2TAA used in Table 4 models. In cases, we now present these summary statistics stratified by median time to diagnosis in Supplemental Table 4 (e.g., below). Effect estimates in Table 4 reflect the covariate-adjusted difference in Log2TAA values, described in the table title and footnote as follows:

P15 (line 246): “Table 4. Linear regression models showing the covariate-adjusted difference in Log2TAA levels associated with ANA-positivity among controls and cases, stratified by time to diagnosisa”

P14 (lines 339-244): “aBetas, calculated through linear regression, show the difference per unit change in Log2TAA values for ANA+ versus ANA- women, adjusting for age, EBNA antibody titers, RF positivity, and current use of hormone therapy. A positive Beta value indicates higher Log2TAA intensity among ANA+ versus women (i.e., greater than zero), whereas a negative Beta value indicates lower Log2TAA intensity among ANA+ women.”

Supplemental materials: “Supplemental Table 4. Mean Log2TAA values among cases, stratified by time to diagnosis and ANA positivity.”

                  Pre-clinical breast cancer cases

Anti-TAA

7+ years to diagnosis

(N=34)

Mean Log2TAA (SE)

ANA+/ANA-

<7 years to diagnosis

(N=35)

Mean Log2TAA (SE)

ANA+/ANA-

PGM3

8.5 (0.67) / 8.0 (0.63)

9.0 (1.0) / 8.3 (0.78)

  1. In your anti-TAA antigen array results, please indicate the ranges for anti-TAA "positivity". If you use fluorescence intensity or log-transformed FI, what are the ranges for a valid value of anti-TAA?  

Thank you for asking. In our methods, we previously wrote about how the value of anti-TAA intensity was derived and used in analyses. We realize that this description could be clearer, and also that we neglected to provide a full accounting in this version of the paper shortened from earlier drafts. We have moved the notation on Log2TAA and added a sentence regarding the range of values. We also added the number of zero values in the Table 2 footnotes and performed a new sensitivity analysis described below. 

P6 (line 125-129): “Normalized values subtracted the log transformed anti-TAA (Log2TAA) median array intensities from individual intensities (median centered normalization) to account for variability between arrays. Valid levels were detected for most antigens identified for analyses below, with low detects reflected by a Log2TAA value of zero. There was no significant difference in zero values by case and ANA status, though the lowest counts were seen among ANA-positive controls (p=0.32; data not shown).”

P11 (line 198-200): aMedian ratio of Log2TAA antibody levels: ANA-positive versus ANA-negative in cases and controls. Those with Log2TAA levels = 0 were excluded from ratio calculations; this included 13 cases (6 ANA+ and 7 ANA-) and 14 controls (4 ANA+ and 10 ANA-) for PRKD1, and 7 ANA- cases and 2 ANA- controls for UCHL1.” 

Although zero values (non-detects) were included in Table 4 models of continuously distributed Log2TAA, we have added a sensitivity analysis to the supplemental materials. This is described in the text as noted below and shown in Supplemental Table 5.

Page 7 (line 157-159): “Zero values were included in regression models of continuously distributed Log2TAA. In sensitivity analyses we imputed values as half the lowest detected values.”

Supplemental materials: Supplemental Table 5. Linear regression models showing the covariate-adjusted difference in mean Log2TAA intensities associated with ANA-positivity among controls and cases, stratified by time to diagnosis with imputed Log2TAA valuesa

Controls

Pre-clinical breast cancer cases

Antigens

No breast cancer diagnosis

N=64

Betab (P-value)

7+ years to diagnosis

N=30

Betab (P-value)

<7 years to diagnosis

N=35

Betab (P-value)

PRKD1

1.14 (0.022)

-1.29 (0.044)

0.71 (0.34)

UCHL1

1.18 (0.029)

ND

ND

ND – not done (no Log2TAA zero values for UCHL1 in cases)

aWhen Log2TAA=0, missing values were imputed as half the value of the lowest value>0. For UCHL1 (2 ANA- controls, Log2TAA_i=3), for PRKD1 (14 controls, Log2TAA_i=1.2; N=3 cases 7+ to diagnosis years Log2TAA_i=1.3; N=10 cases <7 years Log2TAA_i=1.5.

Page 14 (line 240-241) A sensitivity analysis with imputed values for two Log2TAA zero values (PRKD1 and UCHL1) showed no substantial differences (Supplemental Table 5). 

  1. If possible, please use ELISA to valid at least the anti-TTN and anti-PGM3 results.

Retesting specimens is not possible in the context of this pilot study. Validation of these results would be ideal in this sample as well as a larger sample of women from the cohort.

  1. Please clarify what are the association statistic values in Table 4, why, and how some of them are greater than 1?

A linear regression model was used to predict the ratio of log2TAA in ANA+ vs. ANA- women, adjusting for covariates. Please see response to question 6, including the title and footnote edits.  

Reviewer 2 Report

In this study, the authors find that ANA positivity was not broadly associated with higher TAA reactivity, and hence they conclude that ANA are unlikely to be a confounding factor when assessing TAA in relation to breast cancer.

I have a few major concerns:

1. I fail to see why ANA could be a confounding factor in this relation. I think more of ANA and anti-TAA as two sides of the same coin. Are ANA and anti-TAA not just companions? Could the authors perhaps eloborate more on this in the introduction?

2. The authors correctly state, that immunofluorescence on HEp-2 cells is the gold standard for detection of ANA. Despite this, the authors have chosen to screen samples using an ELISA method. Samples positive using ELISA, was then confirmed using immunofluorescence on HEp-2 cells, and if confirmed positive, samples were included. In my opinion, ANA screening should have been performed using immunofluorescence on HEp-2 cells, hence ensuring optimal sensitivity. As the authors state, when using immunofluorescence on HEp-2 cells >150 autoantibodies can be detected. Using ELISA, only a limited repertoire of autoantibodies can be detected.

3. It is correct, that anti-TAA can be detected several years prior to cancer diagnosis. However, in this study, time between serum collection and diagnosis was at an average of 7 years. If time between serum collection and diagnosis had been for a maximum of 2 years, I believe that the authors would have seen a different pattern. Of course, I cannot know this for sure, but still, the authors should address this a major limitation.

Minor comments:

The authors must state precisely which anti-nuclear autoantibodies were tested using ELISA. Section 2.2. Assays.

In the supplementary material, this sentence (table legend) need correction:
'1Baseline history of any self-reported cancer, except for non-melanoma skin cancer; does not include skin cancer, except melanoma.'

A comment: The authors state that litterature on autoimmunity to titin is limited. It is wellknown, that anti-titin is a paraneoplastic autoantibody (thymoma in MG). However, I do agree that the authors findings regarding anti-titin should prompt even more research on the relationship of anti-titin with other cancers and autoimmunity.

Author Response

Reviewer 2

In this study, the authors find that ANA positivity was not broadly associated with higher TAA reactivity, and hence they conclude that ANA are unlikely to be a confounding factor when assessing TAA in relation to breast cancer.

Thank you for your thoughtful review, comments, and questions. We appreciate your feedback and opportunity to improve the manuscript. In our responses and edits, have provided a broader and more nuanced discussion the relationship between anti-TAA and ANA (rather than a focus on potential confounding). We also describe the limitations inherent to the current study design, such as prior methods used for ANA-screening in the base population from which the sample was drawn, and the reason for a lack of samples closer to the time of diagnosis.

I have a few major concerns:

  1. I fail to see why ANA could be a confounding factor in this relation. I think more of ANA and anti-TAA as two sides of the same coin. Are ANA and anti-TAA not just companions? Could the authors perhaps elaborate more on this in the introduction?

Thank you for raising this important point. Confounding is just one possibility, implying a non-causal association that might appear to influence observed relationship of anti-TAA with cancer. At the same time, ANA could be a marker for the tendency towards autoimmunity generally, or perhaps both ANA and anti-TAA are markers of an immune response to a developing tumor or consequential processes leading to the release of self- and tumor-specific antigens. Indeed, anti-TAA appearing years prior to cancer could have several explanations. We have reduced language on confounding and attempted to convey a broader focus in the abstract and introduction, with a more nuanced discussion on the topic.

P2 (lines 21-24): “Autoantibodies to tumor-associated antigens (anti-TAA) are potential biomarkers for breast cancer, but their relationship systemic autoimmunity as ascertained though antinuclear antibodies (ANA) is unknown and warrants consideration given the common occurrence of autoimmunity and autoimmune diseases among women.”

P3 (lines 121-126): “ANA and other more specific autoantibodies often accompany the development of autoimmune diseases [15], many of which are more common in women with autoimmune diseases, such as systemic lupus erythematosus and rheumatoid arthritis. Therefore, a better understanding of the relationship of autoimmunity with tumor-associated autoantibodies is warranted, given the common occurrence of ANA, autoimmune diseases, and breast cancer among women.” ‘

P15 (lines 264-272): “We found that having a positive ANA was not broadly associated with TAA across an array of 171 antigens selected based on prior evidence of associations with breast and other cancers. While largely null, our findings indicate a need for further investigation of TAA and ANA prior to breast cancer diagnosis. The concept of ANA as a potential confounder assumes a non-causal relationship of ANA with both anti-TAA (e.g., in women without cancer) and with breast cancer. Although a few differences in specific anti-TAA levels were seen depending on ANA status among controls, no significant differences remained after adjusting for multiple comparisons. Among cases, however, significantly higher levels of two specific anti-TAA (PGM3 and TTN) were observed in ANA-positive women (Table 2), and in covariate adjusted models, significantly higher anti-TTN levels were seen in women diagnosed within the seven years (Table 4).

  1. The authors correctly state, that immunofluorescence on HEp-2 cells is the gold standard for detection of ANA. Despite this, the authors have chosen to screen samples using an ELISA method. Samples positive using ELISA, was then confirmed using immunofluorescence on HEp-2 cells, and if confirmed positive, samples were included. In my opinion, ANA screening should have been performed using immunofluorescence on HEp-2 cells, hence ensuring optimal sensitivity. As the authors state, when using immunofluorescence on HEp-2 cells >150 autoantibodies can be detected. Using ELISA, only a limited repertoire of autoantibodies can be detected.

The Bead-based assay was likely the most feasible high-throughput method for screening in the original paper, given the extremely large sample size (nearly 10,000 women with self-reported RA). However, we agree this is a limitation of our analysis, as ANA status may be miss-classified due to the initial screening, and some women labeled as ANA negative might in fact have ANA if originally screened by IFA. We have clarified this in the abstract and introduction, while expanding on this limitation in the discussion as underlined below.

P3 (line 28): “The study sample included 145 post-menopausal women with baseline ANA data”

P5 (lines 151-156): “Serum specimens, obtained from a prior study of women with self-reported rheumatoid arthritis screened for ANA, were tested…”

P20 (lines 366-374): “The initial screening ELISA focused on a limited panel of disease-specific autoantigens, with subsequent results confirmed by HEP-2 assays at a level consistent with uses in clinical settings. Not surprisingly, while 15% of the screened sample was deemed ANA-positive, this assay had low sensitivity for the full range of antinuclear antigens. Thus, the true prevalence of ANA may have been higher and represented different specificities, resulting in misclassified ANA-negative women in our analyses, potentially influencing the observed associations. Future research on ANA and anti-TAA would benefit from screening by HEP-2 assay, while at the same time including more data on ANA specificities, titers, and staining patterns, to contextualize results and better understand underlying relationships driving any observed associations.” 

P21 (lines 380-387): “Anti-nuclear antibodies with the dense fine speckled pattern, targeting the LEDGFp75 (lense epitheluium growth factor p75) antigen, are the most common source of ANA in non-autoimmune patients [24], and have been characterized as an “epigenetic reader” that may be upregulated as a stress protein in cancer and inflammatory conditions [25-27]. However, because the available ANA data in the current study were initially screened against a limited disease-specific antigens, the prevalence of this antibody is likely lower among the ANA-positive women in our sample compared to the general population and may be present among the ANA-negative women.”

  1. It is correct, that anti-TAA can be detected several years prior to cancer diagnosis. However, in this study, time between serum collection and diagnosis was at an average of 7 years. If time between serum collection and diagnosis had been for a maximum of 2 years, I believe that the authors would have seen a different pattern. Of course, I cannot know this for sure, but still, the authors should address this a major limitation.

This pilot study design selected a sample nested within a previous biomarker study of women with self-reported RA, starting with all ANA-positive women who became cases during follow-up. Only 4 cases in total were diagnosed within a year following specimen collection, and 16 were within 1-4 years (regardless of ANA status). We hope that our findings can support larger focused studies to include more proximal specimens, perhaps using a nested case-control design, ideally with repeat specimens collected over a broader time prior to diagnosis. We have raised this point more directly in discussion as a limitation and recommendation for future research.  

P3 (line): “Associations of ANA with particular antigens inducing autoimmunity prior to breast cancer warrant further investigation.”

P19 (lines 375-377): “The current study was not intended to investigate the utility of anti-TAA antibodies in breast cancer screening, given the extended time between serum collection and diagnosis (average of 7 years). Results may have been different for samples taken closer to the time of diagnosis.”

Minor comments:

The authors must state precisely which anti-nuclear autoantibodies were tested using ELISA. Section 2.2. Assays.

P5 (line 100-103): “The assay was designed to detect disease-associated antibodies [SSA (SSA 60 and SSA 52), SSB, anti-Sm, sm/RNP, RNP (RNP 68 and RNPA), anti-ribosomal protein, chromatin, anti-dsDNA, centromere, Scl-70, and Jo-1]”

In the supplementary material, this sentence (table legend) need correction:
'1Baseline history of any self-reported cancer, except for non-melanoma skin cancer; does not include skin cancer, except melanoma.'

Thank you for pointing this out; the change has been made and it now reads:1Baseline history of any self-reported cancer, except for non-melanoma skin cancer.”

A comment: The authors state that literature on autoimmunity to titin is limited. It is well known, that anti-titin is a paraneoplastic autoantibody (thymoma in MG). However, I do agree that the authors findings regarding anti-titin should prompt even more research on the relationship of anti-titin with other cancers and autoimmunity.

Agreed – there really should be more research! We now specifically mention thyoma in the text.

Page 16 (line 278-280): “A growing body of research shows antibodies to titin can be associated with the autoimmune muscular disease, myasthenia gravis (closely associated with thyoma) [27]”

Page 18 (lines 313-315): In addition to the example described for TTN, myasthenia gravis, and thyoma described above, cancer-associated autoimmunity has been found to occur across a range of diseases, such as myositis and scleroderma, considered to be a paraneoplastic phenomenon [35].

Reviewer 3 Report

The investigators conducted this exploratory pilot study to examine the relationship between ANA and antibodies to tumor-associated antigens (anti-TAA). In a prior study from the Women's Health Initiative (WHI), elevated serum IgG anti-TAA antibodies were identified to more than 90 antigens weeks before breast cancer diagnosis; this included some reactivity to antigens associated with systemic autoimmune diseases. Because both breast cancer and systemic autoimmunity are common in the general population, the authors raise a question of whether there may be confounding by systemic autoimmunity on the detection of tumor-associated autoantibodies, which provides the foundation for the current study. In this investigation using data and samples from WHI, they examine the relationship of ANA with anti-TAA antibodies (using a panel of 171 antigens) in baseline sera from women who later developed breast cancer and in age and ANA-matched controls who did not have cancer diagnosed during follow up.

The original WHI source population includes 1511 ANA positive women, 85 with breast cancer, and 8473 ANA negative women, 533 with breast cancer. The investigators sampled 145 postmenopausal women from this group: 37 ANA positive women with a first breast cancer during follow up, an age and time to diagnosis matched random sample of 36 ANA negative women who developed breast cancer during follow up, an age and follow up time matched random sample of 35 ANA positive women who did not develop breast cancer, and 37 ANA negative women who did not develop breast or other cancers. Of note, 3 ANA positive breast cancer cases had a cancer history at baseline with other tumor types, and there were other tumors identified during follow up in the ANA positive and ANA negative breast cancer groups. The authors show that anti-TAA levels overall did not vary by ANA positivity in breast cancer cases nor in controls without breast cancer, suggesting that ANA positivity is unlikely to be an important confounder in the use of tumor autoantibodies for research or screening in breast cancer. They did identify 8 specific anti-TAA that were different by ANA status in either breast cancer cases or in controls without breast cancer.

-Throughout the manuscript it is difficult to follow that cases and controls are defined by breast cancer status and not ANA status. In part this is because the abstract is unclear as currently written. I would encourage the authors to revise the abstract but also clarify this throughout the text to improve readability. 

-The source population N in the methods differs slightly from that reported in supplemental table 1: 9973 vs 9984. Could you please clarify?

-Table 1 shows 17% of ANA negative controls had self reported lupus, which seems very high for an ANA negative group. The values are exactly the same as the ANA positive control group though the percentage should be different given the slightly different denominator. Could the authors clarify whether this is correct? It is notable that patients with SLE have been reported to have a lower risk of breast cancer than expected in the general population with a question of whether SLE immune responses exert anti-cancer effects. This may warrant further discussion in terms of how this may affect the interpretation of your results.

-In lines 168-170, the authors discuss time to breast cancer diagnosis in ANA positive cases relative to anti-TAA levels. They mention this was increased for DUSP1. I wonder if this should be DUSP26, and if this should be referred to as ANA negative breast cancer cases instead of ANA positive breast cancer cases based on Table 3. 

-It would be important to address the limitation of breast cancer cases having other tumor types at baseline and during follow up.

-In lines 189-191, it would be better to clarify that this may not be an important confounder for research or screening in breast cancer specifically as your study did not look at other cancer types. The same is true for lines 308-310.

Author Response

Reviewer 3

The investigators conducted this exploratory pilot study to examine the relationship between ANA and antibodies to tumor-associated antigens (anti-TAA). In a prior study from the Women's Health Initiative (WHI), elevated serum IgG anti-TAA antibodies were identified to more than 90 antigens weeks before breast cancer diagnosis; this included some reactivity to antigens associated with systemic autoimmune diseases. Because both breast cancer and systemic autoimmunity are common in the general population, the authors raise a question of whether there may be confounding by systemic autoimmunity on the detection of tumor-associated autoantibodies, which provides the foundation for the current study. In this investigation using data and samples from WHI, they examine the relationship of ANA with anti-TAA antibodies (using a panel of 171 antigens) in baseline sera from women who later developed breast cancer and in age and ANA-matched controls who did not have cancer diagnosed during follow up.

The original WHI source population includes 1511 ANA positive women, 85 with breast cancer, and 8473 ANA negative women, 533 with breast cancer. The investigators sampled 145 postmenopausal women from this group: 37 ANA positive women with a first breast cancer during follow up, an age and time to diagnosis matched random sample of 36 ANA negative women who developed breast cancer during follow up, an age and follow up time matched random sample of 35 ANA positive women who did not develop breast cancer, and 37 ANA negative women who did not develop breast or other cancers. Of note, 3 ANA positive breast cancer cases had a cancer history at baseline with other tumor types, and there were other tumors identified during follow up in the ANA positive and ANA negative breast cancer groups. The authors show that anti-TAA levels overall did not vary by ANA positivity in breast cancer cases nor in controls without breast cancer, suggesting that ANA positivity is unlikely to be an important confounder in the use of tumor autoantibodies for research or screening in breast cancer. They did identify 8 specific anti-TAA that were different by ANA status in either breast cancer cases or in controls without breast cancer.

Thank you for your review and constructive feedback. In addition to much needed changes in the language used to improve clarity on case/control status versus ANA-positivity, we addressed one comment (on observed rates of SLE in relation to cancer) in the table below as well as related notes added to the text. We hope this pilot work will provide some basis for future research on autoimmunity and the development of cancer.

-Throughout the manuscript it is difficult to follow that cases and controls are defined by breast cancer status and not ANA status. In part this is because the abstract is unclear as currently written. I would encourage the authors to revise the abstract but also clarify this throughout the text to improve readability. 

We agree this language was cumbersome and sometimes confusing, so we have attempted to clarify in the abstract and text throughout. For example, operationally defining cases and controls in the abstract and using this terminology more consistently throughout the text:

P3 (lines….): “A total of 37 ANA-positive women who developed breast cancer (i.e., cases; mean time to diagnosis 6.8 years [SE 3.9]) were matched to a random sample of 36 ANA-negative cases by age and time to diagnosis. An age-matched control sample was selected including 35 ANA-positive and 37 ANA-negative women who did not develop breast cancer (i.e., controls; follow-up time ~13 years [SE 3]). We used linear regression to estimate associations of ANA with log-transformed anti-TAA in cases and controls.

 P3 (lines….) “Two anti-TAA were elevated in ANA-positive compared to ANA-negative cases:

-The source population N in the methods differs slightly from that reported in supplemental table 1: 9973 vs 9984. Could you please clarify?

Thanks for catching this error, due to a slight shift of hand on the keyboard so that 8 became 7 and 4 became 3. This has been corrected.

-Table 1 shows 17% of ANA negative controls had self-reported lupus, which seems very high for an ANA negative group. The values are exactly the same as the ANA positive control group though the percentage should be different given the slightly different denominator. Could the authors clarify whether this is correct? It is notable that patients with SLE have been reported to have a lower risk of breast cancer than expected in the general population with a question of whether SLE immune responses exert anti-cancer effects. This may warrant further discussion in terms of how this may affect the interpretation of your results.

This is a very important question and quite intriguing given the literature on SLE and cancer. The numbers are correct. Extracting numbers from the supplemental table 1 and table 1, the following table groups cases and controls by ANA-positivity. The question is whether women with possible RA or SLE could have immune characteristics protective against cancer.  

ANA-positive

ANA-negative

Overall sample

N=1511

N=8473

Anti-CCP abs

12%

7%

Cases

Controls

Cases

Controls

N=37

N=36

N=35

N=37

8%

11%

3%

11%

Overall sample

N=1511

N=8473

Self-reported lupus

13%

6%

Cases

Controls

Cases

Controls

N=37

N=36

N=35

N=37

3%

17%

5%

17%

[We first note that prevalence of anti-CCP antibodies is relatively low in this sample of women with self-reported RA. This is not surprising, since self-report of both RA and SLE is notoriously non-specific (about 1 in 5 at best), so many of the self-reported cases are likely false positives (Validation of self-report of rheumatoid arthritis and systemic lupus erythematosus: The Women's Health Initiative - PubMed (nih.gov).]

In this age matched sample, we find it notable that controls had a higher percent with anti-CCP antibodies (a proxy for RA) and self-reported lupus (regardless of ANA). While intriguing, inferring a potential “protective” associations would be premature, given the small sample size, limitations of the one-time cross-sectional antibody assays and non-specific RA and lupus self-report.

We have added some text to the discussion related to this topic:

P18-19 (lines 328-332): “A growing body of research also suggests potential protective associations of RA and SLE with risk of breast cancer, though the mechanisms are unknown and the evidence is inconsistent [38-40]. The current study design and relatively small sample size precludes a definitive evaluation of ANA and autoimmune diseases in relation to breast cancer, though we note a lower frequency of self-reported lupus in cases compared to controls, regardless of ANA status (Table 1).”

-In lines 168-170, the authors discuss time to breast cancer diagnosis in ANA positive cases relative to anti-TAA levels. They mention this was increased for DUSP1. I wonder if this should be DUSP26, and if this should be referred to as ANA negative breast cancer cases instead of ANA positive breast cancer cases based on Table 3. 

Thank you – we have corrected this typo (page 11).

-It would be important to address the limitation of breast cancer cases having other tumor types at baseline and during follow up.

Also a very good point, which we now address in discussing the limitations of our study sample:

Page 20 (lines 362-366: “To maximize the number of ANA-positive breast cancer cases, our sample included a small number of women with a history of another cancer prior to specimen collection or during follow-up. Sensitivity analyses yielded no substantial differences in TAA/ANA associations after excluding women with DMARDs, CCP antibodies, other cancer, or reported cases of systemic lupus erythematosus.

-In lines 189-191, it would be better to clarify that this may not be an important confounder for research or screening in breast cancer specifically as your study did not look at other cancer types. The same is true for lines 308-310.

Thank you, we have removed this text, integrating more nuanced discussion of confounding in response to reviewer 2, e.g.,

Page 16 (lines 265-269: “The concept of ANA as a potential confounder assumes a non-causal relationship of ANA with both anti-TAA (e.g., in women without cancer) and with breast cancer. Although a few differences in specific anti-TAA levels were seen depending on ANA status among controls (PRKD1 and UHL1), no significant differences remained after adjusting for multiple comparisons.”

Round 2

Reviewer 2 Report

The manuscript is significantly improved, but there are still some minor typos that need to be corrected, ie thyoma. Please proofread thouroghly.